# Naming the Barriers between Anti-CCR5 Therapy, Breast Cancer and Its Microenvironment

**DOI:** 10.3390/ijms232214159

**Published:** 2022-11-16

**Authors:** Elizabeth Brett, Dominik Duscher, Andrea Pagani, Adrien Daigeler, Jonas Kolbenschlag, Markus Hahn

**Affiliations:** 1Department of Plastic, Reconstructive, Hand and Burn Surgery, BG-Trauma Center, Eberhard Karls University Tübingen, 72076 Tübingen, Germany; 2Department of Orthopedics, Traumatology and Hand Surgery, Hospital of Bolzano (SABES), Lorenz-Böhler-Straße 5, 39100 Bolzano, Italy; 3Department of Women's Health, University of Tübingen, 72076 Tübingen, Germany

**Keywords:** breast cancer, tumor microenvironment, CCL5, CCL5/CCR5 axis, mRNA

## Abstract

Breast cancer represents the most common malignancy among women in the world. Although immuno-, chemo- and radiation therapy are widely recognized as the therapeutic trifecta, new strategies in the fight against breast cancer are continually explored. The local microenvironment around the tumor plays a great role in cancer progression and invasion, representing a promising therapeutic target. CCL5 is a potent chemokine with a physiological role of immune cell attraction and has gained particular attention in R&D for breast cancer treatment. Its receptor, CCR5, is a well-known co-factor for HIV entry through the cell membrane. Interestingly, biology research is unusually unified in describing CCL5 as a pro-oncogenic factor, especially in breast cancer. In silico, in vitro and in vivo studies blocking the CCL5/CCR5 axis show cancer cells become less invasive and less malignant, and the extracellular matrices produced are less oncogenic. At present, CCR5 blocking is a mainstay of HIV treatment, but despite its promising role in cancer treatment, CCR5 blocking in breast cancer remains unperformed. This review presents the role of the CCL5/CCR5 axis and its effector mechanisms, and names the most prominent hurdles for the clinical adoption of anti-CCR5 drugs in cancer.

## 1. Background

Breast cancer is one of the most common cancers worldwide, both in developed and less-developed countries. While early breast cancer is generally curable, there are particularly malignant tumors, such as triple-negative breast cancer (TNBC), whose pathophysiology is still unknown, and an effective therapeutic strategy has yet to be identified [1]. In recent years, great progress has been made in the fight against breast cancer, both in surgical and medical treatment. With a multidisciplinary approach, the quality of treatment has significantly improved, and the intensity of local and systemic therapy could be significantly reduced [2].

Growth and invasion of the tumor into surrounding tissues occur at the expense of healthy tissues, which represent the tumor microenvironment (TME). In the past, no particular importance was given to the TME, which was considered harmless and was not thought to have a particular role in breast cancer etiology. The local microenvironment has become a key element only recently, actively participating in cancer progression and representing a fundamental target for new drugs.

At present, surgery, chemotherapy and radiotherapy remain the traditional treatment options for breast cancer, and the approach is mainly based on the severity of the disease, the clinical condition of the patient and his response to treatment. Whereas surgery removes the neoplastic tissue and is the primary treatment for breast tumors in stages I, II or III, radiotherapy destroys the tumor and its microenvironment in a non-specific way.

Among the most popular and traditional chemotherapies, Anthracyclines (Doxorubicin and Epirubicin) and Taxanes (Docetaxel and Paclitaxel) represent key drugs for breast cancer. Even the alkylating agent cyclophosphamide is a mainstay for different tumor types, such as breast, ovarian, breast and blood cancer, retinoblastoma and multiple myeloma. The selective estrogen receptor modulator (SERM) tamoxifen is also a famous drug in oncology and in breast cancer, being able to reduce the development of the tumor mass by blocking the estrogen effects [3,4]. A good percentage of estrogen-sensitive women benefit from therapy with these drugs. For postmenopausal women, the most important drugs are the androgen and anabolic steroid Fluoxymesterone, the aromatase inhibitors Exemestane and Anastrozole and Letrozole, which lowers estrogen levels and is manly used when the positive response after hormone therapy is not sufficient. Other current valuable options to fight breast cancer are represented by the monoclonal antibodies Trastuzumab, Lapatinib and Bevacizumab, most of the time in combination with Paclitaxel [5]. As with all other forms of cancer, prognosis is strongly influenced by the clinical stage at which the cancer is diagnosed. The later the cancer is diagnosed, the more likely it is that the patient will not recover from the disease.

One of the most innovative strategies to fight cancer is represented by Chemokines (CKs), a molecular family of various chemoattractant molecules and cytokines. CKs are involved in homeostasis, angiogenesis, metastasis, immune response, inflammation and chemotaxis. Acting as critical regulators of the immune cell population, CKs are classified into four subfamilies, CXC, CC, CX3C and XC, and are either homeostatic or inflammatory. Homeostatic chemokines are expressed in the lymphoid tissues as a function of immune cell turnover, while inflammatory cytokines are induced upon tissue damage or infection [6]. Within these, the CC motif ligand 5 (CCL5) chemokine (8 kDa) belongs to the CC subfamily and is also known as RANTES (Regulated upon Activation, Normal T-Cell Expressed and Presumably Secreted). By acting as a classical chemotactic cytokine for T cells, eosinophils, basophils and other cells, CCL5 recruits leukocytes to the site of inflammation, induces proliferation of NK cells and is an HIV-suppressive factor released from CD8+ T cells [7]. The receptor with the highest affinity for CCL5 is the CC motif chemokine receptor 5 (CCR5), being mainly expressed in T cells, smooth muscle endothelial cells, epithelial cells and parenchymal cells. The CCL5/CCR5 interaction facilitates inflammation, adhesion and migration of T cells in immune responses. CCR5 is involved in chronic diseases, cancers and COVID-19 infection [8].

In the healthy breast, with the exception of lactating mothers, CCL5 does not exist normally [9]. During lactation, CCL5 is expressed and attracts maternal leukocytes to be included in colostrum [10]. There is a wealth of evidence showing that CCL5 is co-opted in breast cancer [11] and in many other types of tumors, such as pancreatic [12], ovarian [13], prostate [14] and glioma cancer [15]. In vitro and in vivo tests show that blocking or knocking down CCL5/CCR5 is detrimental to tumors such breast cancer and limits metastases [16,17,18]. The possibility of targeting the CCL5/CCR5 axis and inducing an antitumor environment is therefore real but challenging. However, a CCR5 blocker that can be part of cancer therapy has yet to be developed.

Hence, after introducing the importance of the TME in breast cancer and the CCL5/CCR5 axis with its up- and downstream pathways, we highlight the main barriers and hurdles for the clinical adoption of anti-CCR5 therapy in breast cancer. The first hurdle is represented by the great variety of oncological functions of CCR5, which are not yet fully identified. Secondly, the lack of a clear therapeutic window for the application of CCR5 blockers makes the timing of treating the tumor and its microenvironment very complicated. The third and maybe the most difficult hurdle is the uncertainty about which part of the cascade is best to target for the most effective, least problematic pharmaceutical impact. A deep knowledge of all up- and downstream pathways involved in breast cancer is key to understanding which stage of the cascade has to be addressed.

Since the role of CCR5 as a coreceptor for HIV has been completely characterized, a competitive CCR5 blocker named Maraviroc is currently used as effective therapeutical element for HIV patients. However, a simple pivot of Maraviroc to cancer therapy is not feasible, due to possible drug interactions with chemotherapeutic agents.

In the final section, in the wake of mRNA vaccine development, we propose some new pharmacological options for the future.

Altogether, due to its characteristics, the CCL5/CCR5 axis is becoming a valuable therapeutic option. The literature claims that blocking the CCL5/CCR5 connection results in smaller tumors, fewer metastases and longer survival. Despite this promising profile, no CCR5 blocker has been utilized in cancer therapy. Because of the absence of clear publications which explain the barriers and hurdles to the clinical adoption of a CCR5 blocker in cancer therapy, we decided to create such a manuscript to fill this void, mainly through the lens of breast carcinoma.

## 2. Breast Cancer and the Importance of the Tumor Microenvironment

In recent years, the concept of cancer as a single mass of neoplastic cells growing singularly within healthy tissue has changed radically. Breast cancer consists not only of mutated cells, but also of a surrounding altered microenvironment which acts as a critical element for tumor development and progression. Besides breast cancer cells, suppressive immune cells, soluble factors and the extracellular matrix (ECM) act together to promote tumor progression and metastasis. The cellular component of this microenvironment is mainly represented by cancer-associated fibroblasts (CAFs), mesenchymal stromal cells (MSCs), endothelial cells (ECs), pericytes and immune cells [19]. Different molecular alterations and aberrant signaling pathways result in the proliferation of altered stromal cells in the contact zone between the tumor and the surrounding tissue [20,21]. The interaction between cancer cells and the TME is the current focus of cancer research, which aims to identify the specific interplay between them, discover new etiopathogenic pathways and develop new clinical implications.

Our group has always studied breast cancer not as a traditional singular entity but as a real “onco-cellular” system composed of a tumor and its bordering microenvironment. At the contact zone of the primary tumor, there is a delicate interplay between cancer cells and neighboring non-cancer cells. As explained by Provenzano et al. [22], the local microenvironment could be represented by a system of different collagen layers named TACS-1, -2 and -3, which radiate out 90° perpendicular to the contact zone of the tumor. In our recent work [23], we hypothesized that the interaction at the TNBC tumor boundary between adipose stem cells (ASCs) and local fibroblasts (MDA-MB-231 cells) produces high levels of CCL5. This allowed us to demonstrate that resident fibroblasts react to CCL5 production by generating a striated extracellular matrix rich in collagen type VI (Col6a1^−/−^) within the TME.

Because of the sensitivity of the TME to radiotherapy (RT), which induces a loss of hyaluronic acid and alters CAF function within the onco-cellular system [24], we previously highlighted the possibility of targeting the breast cancer and its microenvironment with low doses of RT (5Gy). Our findings showed that irradiated cells in the TNBC microenvironment produce an extracellular matrix which contains lower proportions of oncogenic collagen VI compared to the non-irradiated one [25]. By affecting the TME and inducing an inflammatory reaction, RT impacts the production of collagen, promotes tumor vascularization and leads to cell death.

Because of all these findings, the TME could therefore be represented by distinct layers mainly composed of collagen VI (Col6a1^−/−^) and should be strongly supported by the CCL5-mediated activity of resident fibroblasts. The addition of low doses of adjuvant RT in breast cancer could be a promising therapeutical tool to reduce collagen VI and maybe impact CCL5. This scenario represents an unreported role of radiotherapy in breast cancer and is one of the most hopeful research focuses triggered by our group.

## 3. The Molecule CCL5, the Receptor CCR5 and Its Structural Mutation That Confers HIV Resistance

As introduced before, CCL5 is an extremely powerful chemoattractant with a physiologic role in recruiting immune cells in inflammatory or allergic circumstances [26] CCL5 binds with high affinity to its main receptor CCR5, but also to CCR1, -3, -4, CD44 and GPR75. CCR5 is a seven-transmembrane G-protein-coupled receptor expressed on various cell types, such as T cells, macrophages, dendritic cells, eosinophils and microglia. The interaction between CCR5 and its high-affinity molecules (e.g., CCL5, CCL3, CCL4 and CCL8) results in G protein activation and a following boost of different signal transduction pathways. One of these is represented by NF-kB (Nuclear Factor kappa-light-chain-enhancer), in which CCL5 represents an important target gene [27].

As is widely reported, CCR5 is a critical coreceptor used by HIV in early-stage infection [28]. Blocking CCR5 in HIV patients is a valuable therapeutical option and is relatively innocuous; the CCR5-delta 32 mutation is responsible for HIV resistance and causes the CCR5 extracellular coreceptor to be smaller than usual, and thus defective [29]. The mutation is chiefly prevalent among those descended from Northern Europeans, and Sweden in particular, where the mutation is homozygous in 1% of the population. In total, 10–15% of Europeans have one copy of the mutated gene, which does not confer immunity to HIV, but slows the rate of AIDS development [30]. Despite the number of preclinical and clinical trials and the link drawn between cancer and CCR5 [31], there is no reported, definitive evidence that CCR5 blockers confer cancer resistance on HIV patients.

## 4. The CCL5/CCR5 Axis in Human Diseases and Its Related Downstream and Upstream Pathways

Recently, Zeng et al. summarized the different downstream and upstream pathways correlated with the CCL5/CCR5 axis [32]. The main downstream pathways of CCL5/CCR5 are represented by NF-kB, PI3K/AKT, HIF-1alpha, RAS-ERK-MEK, JAK-STAT and TGF-Beta-Smad. Between these, the activation of AKT and GSK-3Beta through PI3K phosphorylation allows the expression of different downstream proteins, such as Bcl2, Beta-Catenin and Cyclin D, playing a key role in several mechanisms of the cell cycle and cellular apoptosis. In addition, the CCL5/CCR5 axis can support the stability and accumulation of HIF-1alpha, which initiates a cascade of different processes related to angiogenesis and cellular regeneration [33,34,35]. Finally, the activation of the NF-kB pathway through CCL5/CCR5 upregulates the expression levels of Inhibitor of Apoptosis Proteins (IAPs), FLICE-like inhibitory proteins (FLIPs) and matrix metalloproteinase (MMP).

On the other hand, the most important upstream pathways include plasminogen activator inhibitor-1, SOCS-1, Rig1 and Kruppel-like zinc-finger transcription factor 5 (KLF5) [27,36,37,38,39]. Other upstream stimuli of CCL5 are the enhancer of zeste homolog 2 (EZH2), which regulates macrophage-mediated cancer cell progression and migration; HER2 and PTEN, which are associated with cancer progression; and the angiotensin 2 (Ang2), which enhances the transcription of CCL5 and prolongs the immune response [40,41,42,43]. When one or more of these regulators interact incorrectly with the CCL5/CCR5 axis, our body loses its homeostasis and an inflammatory process is established: the tissue fills with common inflammatory molecules such as TNFs, ILs and TGF-Beta, and the inflammatory process initiates a cascade of events leading to disease. In addition to cancer and inflammation, different viral infections, diabetes, Alzheimer’s disease and endometriosis are correlated with the CCL5/CCR5 axis.

As mentioned at the beginning of the manuscript, the relationship between the tumor and its microenvironment is key. Due to the presence of CAFs, MSCs, ECs, different types of immune cells within the TME and the several immunological functions of CCR5, the possibility of regulating the TME through an anti-CCR5 drug is challenging. As thoroughly reported by Jiao [31,44], CCR5 induces cancer cell homing to metastatic sites, enhancing the pro-inflammatory and -metastatic immune phenotype and even DNA repair mechanisms. In the past, much attention had been paid to the combination of CCR5 inhibitors with canonical checkpoint inhibitors (e.g., pembrolizumab). Schlecker et al. [45] highlighted that the synergic effect of these drugs should influence the immunological setting of the TME. Tumor-infiltrating lymphocytes (TILs), myeloid-derived suppressor cells (MDSCs), tumor-associated macrophages (TAMs), innate lymphoid cells (ILCs), Tregs, mesenchymal stem cells (MSCs) and immature dendritic cells could work synergically and contribute to tumor- and TME-induced immunosuppression [46]. The above-mentioned molecules actively express CCR5 and are able to produce CCR5 ligands. MSCs produce CCL3, -4 and 5 and promote metastasis when mixed with breast and colon cancer cells; CD4^+^ Foxp3^+^ Tregs preferentially express CCR5 when compared with CD4^+^ Foxp3^−^ effector T cells; and TAK-779-mediated inhibition reduces Treg migration to tumors, reducing pancreatic tumor size [47] Even the absence of CCR5 ligands is associated with reduced infiltration of antigen-specific T cells and associated metastasis. For example, CCL8 is produced by macrophages in the lungs of mice with metastatic primary tumors. The migration of Tregs toward CCL8 ex vivo is reduced in the presence of Maraviroc. Hence, the treatment of mice with Maraviroc reduced the level of CCR5^+^ Tregs and metastatic tumor burden in the lungs [48]. Another example is given by CCL3, which binds CCR5 and -1, promoting tumorigenesis through recruitment of pro-tumor macrophages into the TME [49]. Because of this, the genetic deletion of CCL3 in macrophages reduces lung metastasis [49]. Numerous other ligands (e.g., EGF, CSF1, HGF, CCL2, CXCR4/CXC1l2 and Tie2) support tumor progression in the TME and are promising therapeutic targets [50]. From a clinical point of view, a CCR5 blockade with anti-CCR5 antibodies can suppress both the growth of melanoma and its TME and MDSC accumulation in mouse tumor tissues. Furthermore, it was shown that Maraviroc reduces MDSC-induced colon cancer metastasis [51].

Above all tumors, breast cancer represents one of the most important and deadliest cancers worldwide. Despite the scarcity of CCL5 in epithelial cells of normal ducts of benign breast lumps, it seems CCL5 is generated during malignant breast transformation [9]. Within the tumor and its microenvironment, the enhanced levels of CCL5 activate the PI3K/AKT/mTOR pathway and lead to cellular proliferation and resistance to apoptosis. The main upregulator of cancer cells is represented by the insulin-like growth 1 (IGF-1) pathway that promotes tumor cell invasion and progression. The enhanced levels of CCL5 lead to high GLUT1 expression on the surface of cancer cells and provide enough energy for the proliferation of breast tumor cells as well as angiogenesis. Even other molecules that increase CCL5 levels, such as IL-6 and HER2-PTEN, contribute to breast cancer development [32].

In addition to CCL5 levels, even CCR5 receptor expression is higher in breast cancer tissues compared to normal tissues. When expressed, CCR5 correlates with increased migratory abilities and cancer invasion [18]. Jiao et al. [44] reported that CCR5+ breast cancer cells are able to form mammospheres and tumors in mice, with high expression of DNA repair pathways. Furthermore, high levels of CCR5 indicate enhanced DNA repair gene levels in response to DNA-damaging agents. Altogether, the interaction of the up- and downstream pathways with the CCL5/CCR5 axis and its impact are under investigation, and the research conducted so far in this direction is up-and-coming.

As mentioned previously, this work shows the main barriers in breast cancer therapy using chemokines. While the first hurdle is represented by the great variety of oncological functions of the CCR5 receptor, which are not yet fully identified, the second barrier concerns the lack of a clear therapeutic window for the application of CCL5/CCR5 blockers. The third and perhaps most difficult hurdle is the uncertainty about which part of the cascade is best to target for the most effective, least problematic pharmaceutical impact.

## 5. Hurdle #1: What Is the Result of Blocking CCR5 in Cancer?

As mentioned before, the CCL5/CCR5 interaction facilitates cancer progression through several different mechanisms. CCL5/CCR5 interaction increases tumor dimensions, induces ECM remodeling, increases cellular migration and metastasis formation, supports cellular stemness and expansion along the tumor borders, confers on cancer cell resistance to therapies, decreases DNA damage, deregulates cellular energetics (metabolic reprogramming), promotes angiogenesis, recruits immune and stromal cells and induces the immunosuppressive polarization of macrophages [27].

There is an unmatched level of evidence supporting the participation of all chemokines other than CCL5 in the construction, development and operation of the primary invasive breast tumor [11]. CCL5 is also ubiquitous across breast cancer cases, being present at stages I, II and III [52], and over 95% of triple-negative breast tumors are CCR5+ [18]. The complete spectrum of roles of CCL5/CCR5 signaling is not known, adding rational uncertainty to the effect of totally blocking the signaling, especially considering evidence showing that high levels of circulating CCL5 are congruent with longer disease-free survival [53]. The authors speculate that CCL5 acts as a double-edged sword—initially fueling tumor development, but also recruiting antitumor cell populations to the zone over time. Such a role could be worth considering before effecting a complete CCR5 blockade. Inhibiting the CCL5/CCR5 signaling cascade does not typically come with drastic side effects or compensatory measures, but the inhibitory agents available for HIV have side effects and negative drug interactions with chemotherapy.

## 6. The Importance of Maraviroc and Other CCR5 Inhibitors

During the study of the CCL5/CCR5 axis in cancer, the main focus was on targeting the interaction through the inhibition of CCR5 with antagonists, with the inhibition of CCL5 expression with neutralizing antibodies or gene silencing and generation of CCL5-knockout mice.

At present, one of the most valuable drugs able to interfere with the CCL5/CCR5 axis is represented by Maraviroc (MVC), the non-peptidic antiretroviral CCR5 blocker used clinically for HIV with the safest profile. MVC is a CCR5 coreceptor antagonist that binds it selectively on the cell membrane, thereby preventing the entry of CCR5-tropic HIV-1 into the host cell. MVC is metabolized by CYP3A4 [54]. As such, it must be considered that chemotherapeutic agents modulating CYP3A4 may represent possible adverse drug reactions [55].

The first preclinical studies by Dorr et al. highlighted a selective and potent activity against all tested CCR5-tropic HIV-viruses. In addition, MVC showed good activity against 200 clinically derived HIV-1 envelope recombinant pseudoviruses and no detectable in vitro cytotoxicity [56]. In an extensive review, C.M Perry [57] explained that Maraviroc has no particular antagonism with PIs (amprenavir, atazanavir, darunavir, indinavir, lopinavir, nelfinavir, ritonavir, saquinavir or tipranavir), NNRTIs (efavirenz, nevirapine or delavirdine) or NRTIs (abacavir, didanosine, emtricitabine, lamivudine, stavudine, tenofovir, zalcitabine or zidovudine). Furthermore, Westby et al. [58] suggested that natural resistance to MVC is rare but there could be some CCR5-tropic HIV-1 variants with decreased drug susceptibility.

As a competitive antagonist of CCR5, MVC inhibits the recruitment of mesenchymal-stromal cells (MSCs), monocytes and some growth factors in order to reduce cancer progression. MVC could also prevent the progression of classic Hodgkin lymphoma (cHL) and indirectly enhance the efficacy of chemotherapy drugs [59]. Other therapeutic strategies based on the use of MVC are currently being tested to fight against some types of cancers, hepatic steatosis, graft-versus-host disease, heart disease, lung disease, type 1 diabetes, rheumatoid arthritis and hemorrhage [32]. Pervaiz et al. is also currently studying the relationship between MVC and breast cancer. In a first preclinical study [60], the authors found that blocking CCR5 reduces the proliferation, colony formation and migration of metastatic breast cancer, inducing apoptosis and G1-phase arrest. Furthermore, MVC treatment inhibits bone metastasis in rats implanted with MDA-MB-231 breast cancer cells [60]. In addition to this, the combination of MVC with other drugs is also promising. Jin et al. [41] recently used MVC and Tocilizumab to demonstrate that IL-6 and CCL5 signaling promote TNBC cell proliferation and migration. A drug combination inhibiting these pathways may therefore be a promising therapy for TNBC patients.

Due to the importance of CCR5, even more drugs have been tested. The CCR5 antagonists with the most success in preclinical studies are Maraviroc, Vicriviroc, Cenicriviroc, TAK-779 and Aplaviroc. Whereas the antiretroviral activity of Aplaviroc was initially promising, in further phase II trials, it showed a dangerous idiosyncratic hepatotoxicity [61,62]. Similarly, despite Vicriviroc showing significant antiretroviral activity and safety in phase II trials, the phase III trial reported high rates of virological failure, and it seemed to block metastasis of human breast cancer xenografts (MDA–MB–231 cells) in immunodeficient mice by inhibiting the cancer cell homing as well as the enhanced killing of these cells through DNA-damaging agents [44,63]. Cenicriviroc, a dual chemokine receptor CCR5/CCR2 inhibitor, is under investigation because of its antiretroviral activity both in vitro and in vivo and its possible use in COVID-19 infections and nonalcoholic steatohepatitis (NASH) with fibrosis. NASH is associated with significant morbidity and mortality due to liver cirrhosis, failure and hepatocellular carcinoma. Cenicriviroc, by antagonizing CCR2/5, blocks fat accumulation and Kupffer cell activation, disrupting monocyte/macrophage recruitment and the hepatic stellate cell activation responsible for fibrogenesis [64,65,66,67]. Finally, a promising therapeutic target in diabetic retinopathy is another dual inhibitor of CCR2/CCR5 named as TAK-779, a quaternary ammonium derivative that reduced retinal vascular leakage in an animal model and Treg infiltration and tumor growth in a pancreatic cancer mouse model [68].

Belonging to a different pharmacological family, Leronlimab is a humanized igG4*k* monoclonal antibody also able to bind CCR5. Adams et al. [69] recently reported some clinical trials testing Leronlimab in metastatic TNBC patients. The phase 1b/2 dose escalation (NCT03838367), Compassionate Use (NCT04313075) and the Basket Study (NCT04504942) were pooled in order to evaluate the drug’s safety and efficacy at 12 months. After the analysis of 28 metastatic TNBC patients, the authors showed that Leronlimab has significant antitumor activity. The clinical trials suggest that metastatic TNBC patients dosed with Leronlimab have a real clinical benefit with improved 1-year progression-free and overall survival and few treatment-emergent adverse events. Finally, after exploring the effect of Leronlimab on circulating tumor-associated cells (TACs) from peripheral blood, the authors revealed that Leronlimab resulted in a drop in circulating TACs in the majority of patients correlating with early therapy response.

## 7. Hurdle #2: What Is the Theerapeutic Window?

Another barrier to the adoption of anti-CCR5 cancer therapy is the lack of a clear therapeutic window in a breast cancer patient. CCL5 is known to chemoattract cells and enhance the invasive ability of cancer cells [70]. A large source of CCL5 in breast cancer is a juxtracrine relationship between adipose-derived stem cells and breast cancer cells [71]. It has been established that blocking either CCL5 or CCR5 immediately after diagnosis would stall cancer cell invasion [72,73,74]. However, the impact of CCL5 on the local stromal fibroblasts could better inform an effective therapeutic window. CCR5 signaling induces local fibroblasts to create linear collagen type VI, leading away from the tumor and into the healthy stroma. The linear, aligned collagen is formally known as a “tumor associated collagen signature” (TACS), a system classing three distinct collagen patterns radiating from the tumor body [22]. Our group showed that linear collagen VI is dependent on CCL5 signaling, as the structural linearity and presence of collagen VI were both significantly decreased upon adding a CCL5 monoclonal antibody to the test conditions [23]. Since the collagen VI isoform is not found in the physiological breast, and since it signifies an increase in malignancy, it is an a priori biomarker for CCR5 signaling.

By the time the CCL5-dependent collagen VI surrounds the tumor, it is likely there are already metastases [23,75]. Moreover, blocking CCR5 after collagen VI has been formed will not destroy the collagen VI. The therapeutic window for blocking CCR5 and collagen VI production has been missed (Figure 1). Therefore, there is suggestive evidence that CCR5 should be blocked as early as possible after the diagnosis.

Data suggest that while the source of the CCL5 is the tumor [23,71], the cells expressing CCR5 are stromal [53]. The pathway to CCL5 production and binding to CCR5 has many moving parts. Is taking a drug meant for HIV and using it in cancer treatment a viable option? If not, what part of the CCR5 pathway should we target for tumor therapy?

## 8. Hurdle #3: What Should Be Targeted and Why?

In drug design, it is critical to understand which stage of the signaling pathway is easiest, least harmful and most effective to alter. In the case of CCL5/CCR5 signaling in breast cancer, four such stages are represented in Figure 2. Briefly, (A) the production of CCL5, (B) free-floating CCL5, (C) substrate/ligand binding or (D) expression of CCR5 are all targetable stages. To affect the cellular source of CCL5 (box A) is to inhibit the cell–cell binding of adipose-derived stem cells and cancer cells, an extremely involved and impractical goal. Meanwhile, the antibodies developed to bind extracellular CCL5 (box B) have found their main application in in vitro research [76]. However, it would take an unreasonable amount of monoclonal antibody given intravenously to reach the breast and have a quantifiable effect. Box C represents the mechanism of action used by most approved drugs, a competitive blocker of the receptor itself. However, considering the advent of mRNA therapy, it becomes fascinating to consider box D, limiting expression of CCR5 at the RNA level.

## 9. mRNA as the New Frontier

In the near future, the use of mRNA could represent the solution for delivering specific proteins or suppressing the expression of key genes within target tissues. In this way, proteins could be coded, or genes could be blocked within pathological tissues, using systemically provided mRNA.

Within the tumor, cancer cells secrete CCL5 and sustain the proliferation of CCR5-positive cells, recruit T-regulatory cells and monocytes, cause osteoclast activation and bone metastasis, neo-angiogenesis and cancer cells dissemination [77]. CCR5 is therefore overexpressed in breast, head, neck, gastric, esophageal, pancreatic and prostate cancer, colorectal carcinoma, melanoma, Hodgkin’s lymphoma, acute lymphocytic leukemia and other tumors [31]. In the clinical setting, higher cytoplasmic CCR5 staining and CCR5 receptor levels correlate with poor prognosis in breast cancer and gastric adenocarcinoma patients. Even elevated levels of its main ligand CCL5 indicate poor prognosis in breast, cervical, prostate, ovarian, gastric and pancreatic cancer and metastatic colorectal carcinoma [31]. The dual ability of CCL5 to provide carcinogenesis or antitumor adjuvanticity depending on the tumor environment appears to be justified by the type of cancer, CCR5 expression and localization of CCL5 expression [78,79]. Because of these considerations, the development of small mRNAs able to bind the CCR5 binding site and induce a transient, autogenous CCR5 block is a challenging task. There is now in-depth knowledge of the CCL5/CCR5 complex, making designing such mRNA possible [80].

The concept of using mRNA to transiently cancel out CCR5 was already explored in 2017 by blocking CCR5 expression with CRISPR-Cas9, shRNA, TALENs, etc. [77]. In the in vitro HIV model of lentiviral CCR5 knockdown, cells had largely normal characteristics, bar a marked resistance to HIV infection [81]. However, such a model with an integrated lentiviral RNA is far from clinical suitability.

Another option for knocking down CCR5 is represented by small interfering RNA (siRNA), which have the ability to decrease invasion and proliferation of cervical cancer cells [17]. The same study used algorithms such as Targetscan, microRNA.org, DIANA and miRwalk to discover possible microRNA (miRNA) targets that regulate expression of CCR5. They discovered miR-107, which is also implicated in invasive breast cancer progression [82].

Based on the evidence, we reason that breast cancer patients could be treated systemically with a CCR5-coding mRNA and reach an antimetastatic effect. However, this could lead to CCR5 desensitization, occurring after prolonged exposure to an agonist, resulting in excessive intracellular CCR5 accumulation in the Golgi apparatus [83]. It is completely unknown how much mRNA would be needed for these approaches, when the treatment would begin and how often it would be administered. Realizing the full set of ramifications of blocking CCR5 by mRNA, or another route, represents an important hurdle for cancer research.

## 10. Conclusions

The TME is a key element in cancer invasion and progression and is the focus for developing new therapeutical strategies. The possibility of targeting a tumor and its TME with chemokines via the CCL5/CCR5 axis is promising, and new therapeutical technologies are under investigation. It is agreed scientifically that blocking CCR5 signaling in breast cancer would have a massive impact on tumor development. The main barriers to this are represented by unknown side effects of blocking CCR5 in a cancer setting, uncertain dose timing and duration and an unclear pharmaceutical mechanism of action for a cancer CCR5 blocker. Considering these three limitations, it makes sense why CCR5 blocking is not yet a mainstay of cancer treatment. mRNA could be the new solution to this old question. Given the sheer wealth of detailed research showing CCL5 as a pro-tumor chemokine, it could be an ideal candidate marker for a highly impactful new mRNA-based drug.

## Figures and Tables

**Figure 1 ijms-23-14159-f001:**
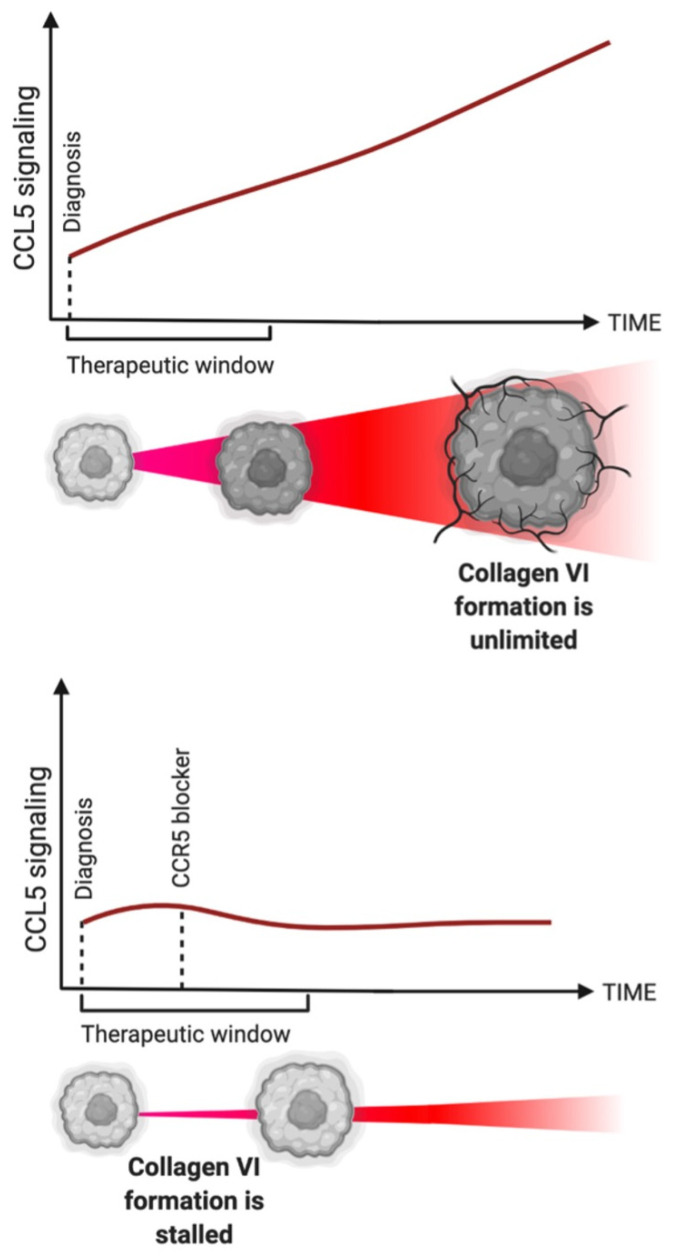
(**Top**): graph showing increasing and unlimited CCL5 signaling over time. Tumor schematic below representing the tumor becoming malignant and creating collagen VI. (**Bottom**): graph showing plateaued CCL5 signaling over time, because of addition of CCR5 blocker within the therapeutic window, i.e., before collagen VI is formed. Tumor schematic below representing the slowed tumor growth due to inhibited collagen VI formation.

**Figure 2 ijms-23-14159-f002:**
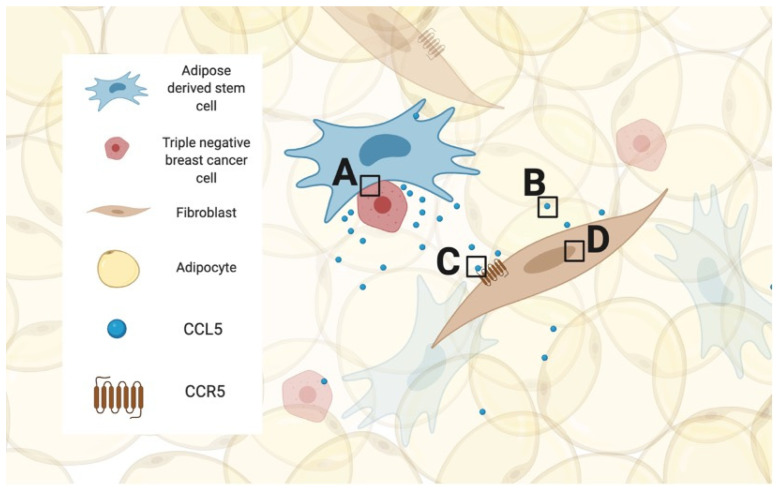
Schematic of options to pharmaceutically manipulate in the CCL5/CCR5 signaling pathway. Figure legend on left. Boxes A–D highlight areas in the cell relationship which represent targetable stages.

## Data Availability

Not applicable.

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
