# Peer review of "Naming the Barriers between Anti-CCR5 Therapy, Breast Cancer and Its Microenvironment"

_ijms, 2022, doi:10.3390/ijms232214159_

Round 1
Reviewer 1 Report
The authors reviewed the possibility of targeting TME with CCL5/CCR5 axis in breast cancer. This review gives the information on the mechanisms of CCL5/CCR5 axis in AIDS and cancers, as well as the most prominent hurdles of anti-CCR5 therapy in cancers. This review is of clinical significance, but there are still several major concerns need to be clarified.
1. TME includes CAFs, MSCs, ECs and all kinds of immune cells. In this paper, immune cells in TME were seldomly discussed. CCR5 mediates physiological functions of immune cells, such as Tregs, MDSCs, DCs, and CCR5 inhibitors exert antitumor effects by avoiding tumor-induced immunosuppression in TME. Therefore, anti-CCR5 therapy regulating the immune cells in TME should be reviewed and discussed thoroughly in the whole text.
2. As the authors mentioned, there are a series of CCR5 antagonists, Maraviroc, Vicriviroc, TAK-779, BMS-813160, and so on. However, only the application of Maraviroc in tumor therapy has been summarized. Other antagonists should also be reviewed. As for Maraviroc, only one reference (ref 47) displays its ability of preventing the progression of lymphoma. Other references on the antitumor efficacy of Maraviroc should be presented, especially for the breast cancer.
3. Are there any references about mRNA-based drugs targeting CCR5 in tumor therapy? These references should be summarized in the last part “mRNA as the new frontier”.
Reviewer 2 Report
Manuscript number: ijms-1967438
Title: Naming the barriers between anti-CCR5 therapy, breast cancer and its microenvironment
General
This review by Brett et. al., presents the role of the CCL5/CCR5 axis, its mechanism, and anti-CCR5 drugs in cancer focusing mainly in breast cancer.
Major Concerns
- Taking into account the title of the review, it completely captures the attention of the readers; however, I suggest delving deeper into each section in order to fully support the title.
- Many sentences have nothing to do with the previous and the next. Many ideas there are not developed only mentioned, but it is necessary to delve deeper. The above is mainly observed in the BACKGROUND section and in the BREAST CANCER AND THE IMPORTANCE OF THE TUMOR MICROENVIRONMENT section. It is necessary that every sentence have a sequence with the next one. I believe that the sentence the lines 141- 142: “In addition, CCL5 is a target gene of NF-kB (Nuclear Factor kappa-light-chain-enhancer of activated B cells)” should be removed from this site and put in another section, it could be at the end of the paragraph. The above is the same case with the sentence in the lines 145-146“CCR5 mediates different signal pathways in response to ligand binding and acts as critical coreceptor for the HIV particles to infect the cell”, I suggest to put this sentences at the beginning of the next paragraph.
- In the lines 244-245, the author are talking about two clinical trials with leronlimab in breast cancer, please add the clinical trial gov identifier, the findings that have been reported and actualize the information at this regard.
- In the subsection of Maraviroc, I suggest to add its mechanism of action and delve deeper in the preclinical studies about this drug and breast cancer. The authors are missing some studies.
- I suggest to delete the last sentence in the abstract section regarding to “we look the recently approved mRNA-based vaccines for inspiration in the pursuit of blocking CCR-5 in breast cancer patients”, because in the last section of the manuscript this is not fully supported.
- Highlight what is new in this review compared to others regarding CCR5 as target in breast cancer therapy.
Minor Concerns
- Please add the complete name of CCR5 and CCL-5 the first time that are mentioned.
- In the second paragraph, lines 44-48, is necessary to delve deeper into current therapies against breast cancer and then do the link with the innovative strategies.
- Line 50: If the authors previously abbreviated the word chemokines, it is necessary to use the abbreviation in the subsequent text.
- In the third paragraph, lines 49-56 is necessary to give an introduction about CCL5, for instance at what kind of subfamily belongs this chemokine and then talk about the CCL5/CCR5 axis and its effects.
- Sentence in the lines 200-201 “CCR5+ breast cancer cells are able to form mammospheres and tumors in mice, with high expression of DNA-repair pathways in contrast to CCR5 cells”. Explain what is highlighted in bold, in contrast to which cells?
- Line 258-259, I suggest to replace the word “degraded” by “metabolized”
Round 2
Reviewer 1 Report
The authors mentioned vicriviroc, cenicriviroc and TAK-779, under the subtitle of “what is the result of blocking CCR5 in cancer”. The application of these CCR5 antagonists in tumor therapy should be summarized, rather than antiretroviral activity.
Author Response
Thank you very much for this suggestion. We updated the section by highlighting the importance of these drugs in tumor therapy by reviewing the original articles of Jiao (2018; 2019) and Soto.
Reviewer 2 Report
The authors made most of the suggestions in the manuscript
Author Response
Thank you very much for reviewing our manuscript.